# Prevalence of Resistance to β-Lactam Antibiotics and *bla* Genes Among Commensal *Haemophilus parainfluenzae* Isolates from Respiratory Microbiota in Poland

**DOI:** 10.3390/microorganisms7100427

**Published:** 2019-10-09

**Authors:** Sylwia Andrzejczuk, Urszula Kosikowska, Edyta Chwiejczak, Dagmara Stępień-Pyśniak, Anna Malm

**Affiliations:** 1Department of Pharmaceutical Microbiology with Laboratory for Microbiological Diagnostics, Medical University of Lublin, Chodźki Str. 1, 20-093 Lublin, Poland; urszula.kosikowska@umlub.pl (U.K.); edyta.chwiejczak@interia.pl (E.C.); anna.malm@umlub.pl (A.M.); 2Sub-Department of Veterinary Prevention and Avian Diseases, Institute of Biological Bases of Animal Diseases, University of Life Sciences of Lublin, Głęboka Str. 30, 20-612 Lublin, Poland; dagmara.stepien@up.lublin.pl

**Keywords:** *Haemophilus parainfluenzae*, beta-lactam resistance, *bla* genes, *ftsI*

## Abstract

(1) Background: Beta-lactams are the most frequently used antimicrobials, and are the first-line drugs in many infectious diseases, e.g., pneumonia, otitis media. Due to this fact, various bacteria have developed resistance to this group of drugs. (2) Methods: Eighty-seven *Haemophilus parainfluenzae* isolates were obtained from adults 18–70 years old in eastern Poland. The presence of 10 *bla* genes and 2 substitutions in *ftsI* reported as the most frequent in *H. parainfluenzae* were analyzed. (3) Results: Among 57 beta-lactam-resistant isolates, 63.2% encoded *bla* genes; *bla_TEM-1_* predominated (54.4%), followed by *bla_OXA_* (19.3%), *bla_DHA_* (12.3%), *bla_SHV_* (10.5%), *bla_GES_* (7.0%), *bla_CMY_* (5.3%), *bla_VEB_* (1.8%) and *bla_ROB-1_* (1.8%). Lys-526 was the most common substitution in *ftsI* gene. The resistance genotypes were as follows: gBLNAS (17.5%), low-gBLNAR I (1.8%), low-gBLNAR II (1.8%), gBLNAR II (15.8%), gBLPAS (15.8%), gBLPAR (19.3%), gBLPBS I (8.8%) and gBLPBS II (1.8%); (4) Conclusions: This has been the first study to report on the high diversity of *bla* genes in *H. parainfluenzae* isolates in Poland. High sensitivity and specificity of benzylpenicillin test, as well as PCR of *bla* genes were shown, indicating that these methods may be useful as tools for the rapid screening of beta-lactamase prevalence and resistance to beta-lactams among *H. parainfluenzae* isolated from respiratory microbiota.

## 1. Introduction

Beta-lactam antibiotics and especially penicillins, are the most frequently used group of antimicrobials in the European Union. In Poland, their consumption in the community for systemic use was about 10.0 defined daily doses (DDD) per 1000 inhabitants per day in 2017, while in Germany, Sweden, and Austria it was in the range of 5.0–7.2 DDD per 1000 inhabitants per day [1]. Beta-lactams (e.g., amoxicillin, second- or third-generation cephalosporins) are of great importance in the treatment of many infections, including those caused by haemophilic rods (*Haemophilus* spp., *Pasteurellaceae* family). They are the first-line drugs in the outpatient treatment of pneumonia, with suspicion of aspiration pneumonia in adults, and in treatment of acute bacterial rhinosinusitis, acute otitis media, and bacterial meningitis [2]. Amoxicillin is recommended for the treatment of non-beta-lactamase-producing *Haemophilus influenzae* infections, while a second- or third-generation cephalosporins or beta-lactams combined with beta-lactamase inhibitors (e.g., amoxicillin–clavulanate) are preferred for infections caused by beta-lactamase-producing bacteria [2,3,4].

Due to the very frequent and widespread use of beta-lactams, the development of resistance to this group of drugs has been observed also among isolates of *H. influenzae* and *Haemophilus parainfluenzae* [5,6,7,8,9,10,11,12,13,14,15,16,17,18,19,20,21,22,23]. The mechanism of action of beta-lactam antibiotics is the inhibition of transpeptidases, mainly peptidoglycan and penicillin-binding proteins 1–7/8 (PBPs). In *H. influenzae*, five different PBPs have been frequently reported, and PBP3 is the most correlated with resistance to beta-lactams [5,18]. Bacteria have developed a number of mechanisms of resistance to beta-lactam agents as follows: (1) production of specific enzymes that hydrolyze antibiotic molecules, (2) production of PBP proteins with a reduced affinity for beta-lactam antibiotics, (3) lowering the permeability of cellular shields or (4) active efflux of drug outside the bacterial cell [13,16,21,24].

The most common resistance mechanism to beta-lactams is the production of beta-lactamases, specific enzymes with hydrolytic properties, encoded by *bla* genes located on small plasmids [25,26,27,28]. Numerous authors have reported the presence of a variety of beta-lactamases in *H. influenzae*, with TEM-1 or ROB-1 as the most predominant [22]. Both belong to Ambler’s class A serine beta-lactamases linked with ampicillin resistance, effectively inhibited by beta-lactamase inhibitors (e.g., clavulanate and sulbactam) [5,29].

The mechanism of resistance, consisting of the production of PBP proteins with a low affinity to beta-lactams, is a two-way process. The first strategy is the modification of natural PBP proteins, caused by numerous mutations in the *pbp* genes or the acquisition of altered fragments of homologous genes from other microorganisms and their substitution. This results in maintenance of the catalytic functions in the production of peptidoglycan, with a simultaneous lack or reduction of affinity for the antibiotic [16]. The second strategy is to acquire a complete foreign gene encoding a PBP protein that is devoid of targets capable of interacting with beta-lactam antibiotic molecules [5,24]. According to literature, a third strategy is possible, consisting of the simultaneous presence of both mechanisms of resistance in the same strain, resulting in a strain that can produce beta-lactamases and as well as presenting modifications in the structure of PBP proteins [5,16].

The aim of this study was to determine the in vitro mechanisms of resistance to beta-lactam antibiotics in *H. parainfluenzae* isolates selected from respiratory tract microbiota of adults in eastern Poland. To achieve this, a simple PCR amplification method was used to detect ten different *bla* genes, the *ftsI* (*pbp3*) gene, and commonly reported Thr-385 and Lys-526 substitutions in *ftsI* gene. This report also describes the impact of beta-lactam resistance genotypes on antibiotic susceptibility of *H. parainfluenzae* clinical isolates.

## 2. Materials and Methods

### 2.1. Bacterial Isolates

A total of 87 *H. parainfluenzae* isolates were obtained from throat or nasopharyngeal swabs from adults (18–70 years old), both healthy individuals and patients with chronic diseases (lung cancer, chronic lymphocytic leukaemia) in eastern Poland between 2013 and 2015 (the Medical University of Lublin, Poland Bioethical Commission No. KE-0254/75/2011 28 April 2011 and No. KE-0254/59/2016 25 February, 2016). The following reference strains of the *Haemophilus* genus from the American Type Culture Collection (ATCC) were also used as positive controls: *H. influenzae* ATCC 49247 (beta-lactamase-negative ampicillin-resistant strain, BLNAR) and *Escherichia coli* ATCC 35218 (*bla*_TEM-1_-positive strain). As a negative control, *E. coli* ATCC 25922 (extended-spectrum beta-lactamase-negative strain, ESBL-negative) strain was used.

### 2.2. Culture and Identification

Isolates were stored as a frozen stock in trypticasein soy broth (TSB, Biocorp, Warsaw, Poland) supplemented with *Haemophilus* test medium supplement (HTMS, Oxoid, Hampshire, Great Britain) with addition of 30% (v/v) glycerol at −70 ± 2 °C until its use. Bacteria were then re-cultured by applying the frozen stock to a chocolate agar (BioMérieux, Craponne, France) and incubated for 24 h at 35 °C in microaerophilic (5–10% CO_2_, 80–90% N_2_, 5–10% O_2_, Generbag microaer, BioMérieux, Craponne, France) conditions. *Haemophilus* spp. isolates were then identified by colony morphology, Gram-staining, and identified to the species level by API NH microtests (BioMérieux, Craponne, France) and by the Ultraflextreme Matrix Assisted Laser Desorption Ionization Time of Flight Mass Spectrometry (Bruker Daltonics, Bremen, Germany) (MALDI-TOF MS) with MALDI-Biotyper 3.0 software (Bruker Daltonics, Bremen, Germany) according to the procedure described earlier [30]. The correctness and reliability of the abovementioned identifications were expressed in the form of a point indicator, as presented previously [31]. Only *H. parainfluenzae* isolates with identification scores >1.999 on the basis of protein profile were taken for further analysis.

### 2.3. Antimicrobial Susceptibility Testing

The isolates grown overnight on chocolate agar (HAEM, BioMérieux, Craponne, France) were resuspended in TSB + HTMS medium. Cell concentration in broth medium was determined using a densitometer (BioMérieux, Craponne, France). Bacterial suspensions with a density of 0.5 (1.5 × 10^8^ colony forming units CFU/mL) were used according to the McFarland standard and then incubated overnight at 35 °C in microaerophilic atmosphere. Each isolate was tested for beta-lactamase activity by the Pen reaction on API NH strip, and tested with a benzylpenicillin disk (1 U, Becton Dickinson, Franklin Lakes, NJ, USA) and the cefinase disk assay (Becton Dickinson, Franklin Lakes, NJ, USA). According to the benzylpenicillin disk screening test [32], isolates with zone diameter ≥ 12 mm were reported as susceptible to any beta-lactam agents for which clinical breakpoints were available (all beta-lactam resistance mechanisms excluded). Isolates with zone diameter < 12 mm were suspected to produce beta-lactamases and/or to have PBP3 mutations, and needed further analysis. As for ampicillin and amoxicillin without beta-lactamase inhibitors, if beta-lactamase was positive, the isolate was reported as resistant, and if beta-lactamase was negative, then as susceptible according to the clinical breakpoints. As for other beta-lactams, except cefepime, cefpodoxime, and imipenem, the isolate was reported according to the clinical breakpoints for the agent in question. Additionally, for the mentioned agents, if resistant according to both screen and agent disk diffusion test, the isolate was reported as resistant. If isolate was resistant according to the screen test and susceptible according to the agent disk diffusion test, the minimal inhibitory concentration (MIC) of the agent was determined and interpreted according to latest breakpoints. As regards the cefinase test, all *H. parainfluenzae* isolates were classified according to the color change of the nitrocefin disc after 2–3 min as either beta-lactamase-positive isolates (cef^+^, change of the yellow color to pink) and beta-lactamase-negative (cef^−^, no change of color). In a subsequent step, the susceptibility to the following beta-lactam antibiotics: ampicillin (2 µg), amoxicillin–clavulanate (2/1 µg), ampicillin-sulbactam (10/10 µg), cefuroxime (30 µg), cefotaxime (5 µg), imipenem (10 µg), and meropenem (10 µg) was determined using the Kirby–Bauer disk-diffusion method [33], using Mueller–Hinton agar medium with 5% horse blood with the addition of 20 mg/L NAD (MHF, BioMérieux, Craponne, France). Moreover, MIC alues of ampicillin (MIC_Am_) for ampicillin-resistant isolates were determined by the E-test method using E-test strips (BioMérieux, Craponne, France) at a concentration gradient of 0.016–256 mg/L. Diameters of bacterial growth inhibition zones were measured using the Interscience Scan^®^ 1200 version 8.0.3.0 (Interscience, St Nom la Bretèche, France); the results were interpreted according to the latest European Committee on Antimicrobial Susceptibility Testing (EUCAST) 2019 criteria [32].

### 2.4. DNA Extraction

Genomic DNA was extracted using a modified boiling method [34]. The culture was transferred into sterile 1.5 mL Eppendorf tubes, followed by centrifugation for 10 min at 14,000 rpm. The obtained supernatant was decanted from the precipitate. The pellet was resuspended in 200 µL of sterile deionized water and centrifuged again (14,000 rpm for 5 min). Next, the obtained supernatant was recovered from the pellet and the pellets were once more suspended in 100 μL of sterile deionized water. The contents of the tubes were thoroughly mixed by vortexing for 10 s at 13,000 rpm and then incubated for 10 min on a thermoblock (Biometra, Göttingen, Germany) at 100 °C. After cooling, samples were subjected into centrifugation for 10 min at 14,000 rpm. The supernatant, containing isolated DNA, was transferred to new sterile Eppendorf tubes, which were reheated for 10 min at 100 °C. Extracted DNA was frozen to −70 °C for further analysis.

### 2.5. Amplification Experiments and Gene Detection

Determination of the presence of ten beta-lactamase *bla* genes: *bla*_TEM-1_, *bla*_GES_, *bla*_OXA_, *bla*_VEB_, *bla*_CTX-M-1_, *bla*_SHV_, *bla*_CMY_, *bla*_DHA_, *bla*_PER_, and *bla*_ROB-1_ (also called further: TEM-1, GES, OXA, VEB, CTX-M-1, SHV, CMY, DHA, PER, and ROB-1, respectively) was carried out for all *H. parainfluenzae* isolates. PCR amplification from genomic DNA was performed for parts of the abovementioned genes. Table 1 presents previously published sequences of the oligonucleotides (Novazym, Poznan, Poland). The PCR cycling conditions were 30 cycles of the following: 95 °C for 30 s, 46 °C for 60 s, and 72 °C for 30 s. All reactions were carried out using the AmpliMIX HiFi (Novazym, Poznan, Poland) in a total volume of 25 µL containing: 0.1 µL of 0.5 U HiFi *Taq* polymerase DNA, 2.5 µL of reaction buffer (10×) pH 8.6, 0.75 µL of 2 mM dNTPs mix, 1 µL of each 0.6 µM primer, followed by electrophoresis in 1.5% agarose gel (Sigma-Aldrich, Saint Louis, MO, USA). Each reaction included 2 µL of DNA templates from individual *H. parainfluenzae* isolates.

Determination of presence of the *ftsI* (*pbp3*) gene encoding the transpeptidase domain of PBP3 protein was performed by PCR reaction according to Touati et al. [18]. Additionally, the amplifications with primers complementary to the part of *ftsI* gene within which amino acid substitutions (Table 1) resulting in resistance to beta-lactam antibiotics have been commonly reported used *pbp3*-BLN (Thr-385 and Lys-526 substitutions) and *pbp3*-INT (Lys-526 substitution) [19].

Based on the results obtained by the beta-lactams susceptibility testing, drug resistance phenotypes and genotypes of drug resistance to beta-lactam antibiotics were classified as follows (Table 2).

### 2.6. Statistical Analysis

The data processing and analysis were performed using GraphPad InStat 3.00 (GraphPad Software, San Diego, CA, USA). Fisher’s exact test was used to calculate the 95% confidence interval ranges (95% CI) and the relative risk (RR). A *p* value of <0.05 was considered statistically significant. The following indexes were calculated for the conventional phenotypic methods used to detect beta-lactamases in relation to PCR samples: sensitivity, specificity, positive predictive value (PPV), and negative predictive value (NPV). Sensitivity (defined as the percentage of true positives) was calculated as the proportion of *H. parainfluenzae* isolates that tested positive among all isolates tested that actually produced beta-lactamase, whereas specificity (defined as the percentage of true negatives) was calculated as the proportion of isolates that tested negative among all isolates that actually did not produce beta-lactamase. PPV and NPV allowed a clinical perspective to be obtained of how likely production of beta-lactamase was in comparison to PCR results among *H. parainfluenzae* isolates tested. Positive predictive value was the probability that, following a positive test result, an individual bacterial isolate would truly produce that specific beta-lactamase, while negative predictive value was the probability that following a negative test result, that individual bacterial isolate would truly not produce that specific enzyme.

## 3. Results

### 3.1. Antimicrobial Susceptibility Patterns Among H.parainfluenzae Isolates

Among all tested *H. parainfluenzae* isolates, 34.5% (30/87) were sensitive to every used beta-lactam antibiotic and were excluded from further analysis, and 65.5% (57/87) were resistant to beta-lactams. Detailed percentages were as follows: 36.8% (32/87) for ampicillin, 37.9% (33/87) and 35.6% (31/87) to cefotaxime and cefuroxime, respectively (Figure 1).

Figure 2 shows that among all isolates, 36.8% (32/87) were ampicillin-resistant; for 10 of them MIC_Am_ values ranged from 0.5 to 1.0 mg/L (sensitive), for 15 from 1.5 to 3.0 mg/L (susceptible, increased exposure, formerly intermediate), and 7 were ≥6.0 mg/L (resistant).

### 3.2. The incidence of β-Lactamase-Positive Isolates According to Phenotypic Methods

In only 5.7% (5/87) of *H. parainfluenzae* isolates was the cefinase test. In 91.9% (80/87) of isolates, the benzylpenicillin zone diameter of growth inhibition was <12 mm, and those isolates were suspected to produce beta-lactamases and/or have PBP3 mutations. Additionally, amoxicillin–clavulanate-resistant (2/1 µg) isolates were reported in 2.3% (2/87) cases. According to the results of beta-lactamase synthesis (on the basis of three phenotypic methods as follows: cefinase test, penicillinase production in API NH microtest, and amoxicillin–clavulanate 2/1 µg susceptibility test), altogether 13.8% (12/87) isolates were phenotypically able to synthesize beta-lactamases, while 81.6% (71/87) did not produce any beta-lactamase. For 4.6% (4/87) of the *H. parainfluenzae* isolates, at least one of the tests used (PEN in API NH strip) gave uncertain or ambiguous results (e.g., a weak reaction), which was statistically significant (*p* < 0.0001, 95% CI 0.1031–0.2986, RR = 0.1754).

Statistical diagnostic values of the conventional phenotypic methods used to determine beta-lactamase production among *H. parainfluenzae* isolates in comparison to PCR amplification results (*bla* gene identification) are shown in Table 3. The sensitivity of the standard diagnostic methods ranged from 51.33% for amoxicillin–clavulanate 2/1 μg susceptibility test to 90.63% for benzylpenicillin screen results. Specificity of those methods was 100%, except for the penicillinase test (API NH) which had a value of 96.43%.

### 3.3. Prevalence of β-Lactamase Genes

Figure 3 presents the beta-lactamase gene distribution among *H. parainfluenzae* isolates according to PCR analysis. Of the 87 *H. parainfluenzae* isolates tested, 65.5% (57/87) were resistant to beta-lactams, among which 63.2% (36/57) were beta-lactamase-gene-positive (*bla*+): 54.4% (31/57) were TEM-1^+^, followed by 19.3% (11/57) OXA^+^, 12.3% (7/57) DHA^+^, and 10.5% (6/57) SHV^+^ beta-lactamase genes. None of the tested isolates encoded *bla*_PER_ or *bla*_CTX-M-1_ genes.

### 3.4. Detection of ftsI Gene and Amino Acid Substitutions.

Among all tested *H. parainfluenzae* isolates, 37.9% (33/87) were *ftsI* positive (*ftsI*^+^), among which 78.8% (26/33) were beta-lactam-resistant and 21.2% (7/33) were sensitive (Table 3). Moreover, of the beta-lactam-resistant *H. parainfluenzae* isolates, 61.5% (16/26) of the *ftsI*^+^ isolates were beta-lactamase-positive. The most frequent amino acid change was at the Lys-526 position of *ftsI*, found in 93.8% (15/16) of *ftsI*^+^
*bla*^+^ isolates. The Thr-385 substitution in *ftsI* was found in 12.5% (2/16) *ftsI*^+^
*bla*^+^ isolates. All 38.5% (10/26) *ftsI*^+^ beta-lactamase-negative (*bla*^−^) *H. parainfluenzae* isolates resistant to beta-lactams had only the Lys-526 substitution in the *ftsI* gene (Table 4).

### 3.5. Relationship between Susceptibility to β-Lactam Antibiotics and Resistance Genes

According to the genotype classification of 57 beta-lactam-resistant *H. parainfluenzae* isolates, 17.5% (10/57) were gBLNAS, 1.8% (1/57) was low-gBLNAR I and 1.8% (1/57) low-gBLNAR II, 15.8% (9/57) were gBLNAR II, 15.8% (9/57) were gBLPAS, 19.3% (11/57) were gBLPAR, 8.8% (5/57) were gBLPBS I and 1.8% (1/57) was gBLPBS II, and 15.8% (9/57) were gBLPACR I and 1.8% (1/57) was gBLPACR II (Table 2).

Within all beta-lactam-resistant isolates, 36.8% (21/57) were beta-lactamase-negative, classified into gBLNAS, low-gBLNAR, and gBLNAR genotypes. The gBLNAS isolates were resistant to one (Ctx/Ipm) or two (Cxm Ctx/Ctx Mem/Cxm Mem) antibiotics simultaneously, and the resistance to cefotaxime found in 33.3% (7/21) of isolates was predominant. Two low-gBLNAR isolates were resistant only to ampicillin. One of them—the W1HC isolate—had a MIC_Am_ value of 0.8 mg/L (susceptible according to EUCAST) and the Lys-526 substitution in *ftsI*. The second one—the IM18GB isolate—had a MIC_Am_ value of 2.0 mg/L susceptible, increased exposure (formerly intermediate according to EUCAST), did not harbor the *ftsI* gene or any amino acid substitutions, and was classified into the low-gBLNAR I genotype.

All 14.0% (8/57) tested gBLNAR isolates were 100% resistant to ampicillin and cefuroxime, 88.9% resistant to cefotaxime, seven of them had an Am Cxm Ctx resistance pattern, and one had an Am Sam Cxm Ctx pattern. The MIC_Am_ values ranged from 2.0 to 8.0 mg/L (two had MIC > 4.0 mg/L), and one 39CU isolate was cefinase-positive (BLPAR phenotype). All gBLNAR isolates were classified into group II with an amino acid substitution at the Lys-526 position in the *ftsI* gene.

Of the 36 phenotypically beta-lactamase-positive isolates, 41.7% (15/36) were susceptible to ampicillin and were classified into the BLPAS phenotype. Among them, resistance to cefuroxime was predominant, observed in 73.3% (11/15) of isolates, followed by cefotaxime, found in 53.3% (8/15) of cases. Four of them were resistant to two beta-lactam antibiotics (Cxm Ctx or Cxm Ipm), one was resistant to three antimicrobials (Cxm Ctx Mem) and one was resistant to four antibiotics, presenting a Cxm Ctx Ipm Mem resistance pattern. In 40% (6/15) of BLPAS isolates, amino acid substitutions at the Lys-526 position of the *ftsI* gene were found, in one isolate, the Thr-385 substitution was observed. Due to that, we classified these isolates into the gBLPBS genotype, divided further into two groups: I with the Lys-526 substitution and II with the Thr-385 substitution. Eleven (30.5%) of the 36 beta-lactamase-positive isolates were BLPAR, among which two were cefinase-positive. All BLPAR isolates were ampicillin-resistant, one had MIC_Am_ = 0.75 mg/L (sensitive according to EUCAST), six had MIC_Am_ values from 1.0 to 3.0 mg/L (susceptible, increased exposure according to EUCAST), and four had MIC_Am_ = 6.0 mg/L (resistant according to EUCAST). According to PCR results, the assigned phenotype of one isolate was not covered by the genotype due to the presence of an amino acid substitution at the Lys-526 position in the *ftsI* gene, thus, the W5HD isolate was classified into the gBLPACR I genotype. Similarly, two cefinase-positive BLPBR isolates, one resistant to four (Am AmC Cxm Ctx), and one resistant to six (Am AmC Cxm Ctx Ipm Mem) beta-lactam antibiotics, with Lys-526 substitutions in the *ftsI* gene, were classified into the gBLPACR I genotype.

## 4. Discussion

### 4.1. Resistance to Beta-Lactams among H. parainfluenzae 

Increasing resistance to antimicrobial agents is a global phenomenon nowadays, commonly occurring among haemophilic bacteria, including the rare etiological factor of such infections as caused by *H. parainfluenzae* [13,14,15,21]. This may have a huge impact on therapeutic treatment for infections caused by these bacteria. The importance of this issue has been evidenced by an increasing number of reports on beta-lactamase-producing strains and the weakening activity of beta-lactam antibiotics against *H. parainfluenzae* [7,13,14,16,21]. Since 2013 only a few publications have been published in the PubMed database about *H. parainfluenzae* resistance to beta-lactam antibiotics and its molecular mechanisms [7,13,16,21], despite the fact that it belongs to the fastidious Gram-negative bacterial group of the *Haemophilus–Aggregatibacter–Cardiobacterium–Eikenella–Kingella* genus (HACEK). That means *H. parainfluenzae* is an etiological factor of many documented chronic or recurrent infections also occurring within the respiratory system [2,18]. This was the first study to report the high diversity of beta-lactamase genes in *H. parainfluenzae* isolates from respiratory tract microbiota in eastern Poland.

### 4.2. The Susceptibility to Beta-Lactam Antibiotics among H. parainfluenzae Isolates 

The susceptibility of *H. influenzae*, and similarly *H. parainfluenzae*, to beta-lactam antibiotics is mainly determined by ampicillin susceptibility results. The resistance phenotypes for these bacteria are divided into many groups and subgroups [9,10,11,12,19,24,42]. In *H. influenzae* clinical strains, resistance to ampicillin resulting from the bacterial ability to synthesize beta-lactamases is usually detected on the basis of MIC values for ampicillin and amoxicillin with clavulanic acid above the recommended values as the cut-off point for resistant strains. According to the actual EUCAST recommendations for amoxicillin–clavulanate, the area of technical uncertainty (ATU) is relevant only if the benzylpenicillin 1 U disk screen is positive [32].

In our study, among all tested isolates, 34.5% were sensitive and 65.5% were resistant to tested beta-lactams, with resistance to ampicillin, cefuroxime, and cefotaxime predominating and found in 36.8%, 37.9%, and 35.6% of isolates, respectively. Other researchers have also reported a high extent of resistance among *H. parainfluenzae* isolates [7,14,15,21,23]. Tinguely et al. [7] described an extensively drug-resistant (XDR) *H. parainfluenzae* isolate resistant to almost all tested beta-lactams, comprising: ampicillin, amoxicillin, amoxicillin–clavulanate, cefuroxime, ceftriaxone, cefotaxime, and cefepime. Abotsi et al. [14], in studies on *H. parainfluenzae* isolated from sputum of patients with pneumonia, also demonstrated its resistance to fluoroquinolones and telithromycin. In turn, the phenotypic resistance to beta-lactam antibiotics, azithromycin, and trimethoprim–sulfamethoxazole was demonstrated in *H. parainfluenzae* by Kosikowska et al. [15]. The research of these authors was devoted to microorganisms isolated from the airways of patients suffering from lung cancer resistant simultaneously to five, six, or eight drugs from different therapeutic groups, including beta-lactams.

### 4.3. Conventional Methods for Detection of Beta-Lactamase in Haemophili 

Out of the three methods available for the detection of beta-lactamase in haemophili, the cephalosporin (nitrocefin) method is the most reliable and recommended in the case of haemophilic bacteria [32,43]. According to a manufacturer of the API NH strips, a positive penicillinase test indicates the presence of a penicillinase, which prohibits the use of penicillins (penicillin G, amino-, carboxy-, and ureidopenicillins). Additionally, a susceptibility test is required for the other beta-lactams. However, according to EUCAST, the benzylpenicillin 1 U disk screen test can be used to exclude beta-lactam resistance mechanisms [32,42]. When screening is negative, all beta-lactam agents for which clinical breakpoints are available can be reported sensitive without further testing. This result excludes both beta-lactamase production and other beta-lactam resistance mechanisms. All positive screen results indicate a possibility of both resistance mechanisms—beta-lactamase production and/or PBP3 mutations, as was explained in the Materials and Methods section based on the EUCAST flow chart [32]. At the same time, additional results should be taken into account, e.g., ampicillin and amoxicillin susceptibility [32,42]. However, phenotypic methods are sometimes insufficient to visually detect these enzymes, or the data obtained are ambiguous and difficult to interpret [20,43].

On the basis of the three phenotypic methods used in our study, 13.8% of isolates were able to phenotypically synthesize beta-lactamases, and in 4.6% of isolates, at least one of the tests used (PEN in API NH strip) gave a less precise and difficult to interpret (weak color change) result, which was statistically significant (*p* < 0.0001). Moreover, statistical diagnostic values of these methods among *H. parainfluenzae* isolates in comparison to PCR amplification results (*bla* gene identification) showed that the test of susceptibility to amoxicillin–clavulanate was the most sensitive, while the benzylpenicillin screen test was the most specific for *H. parainfluenzae* clinical isolates.

### 4.4. PCR Amplification as Rapid Detection Method of Beta-Lactamase Production 

Many researchers have questioned the utility of conventional phenotypic methods in microbial diagnostics, requiring at least 2–3 days to fully detect and determine drug susceptibility [20,36,43,44]. They indicate treatment failure, poor outcome, and development of beta-lactam resistance among *Haemophilus* spp. strains as consequences of negative identification. It has been suggested that more specific and sensitive methods should be used, including PCR amplification or real-time PCR for quantification of *Haemophilus* spp. strains and simpler, more rapid and reliable detection of beta-lactamase production [7,13,14,17,18,19,21]. For example, [19] revealed 92.9% and 91.8% sensitivity and specificity, respectively, of the PCR amplification method used to detect ampicillin-resistant intermediate *H. influenzae* strains, and 100% sensitivity and specificity of primers used to identify TEM-1 beta-lactamase compared to the conventional phenotypic methods.

### 4.5. Prevalence of Beta-Lactamase Type TEM-1 and ROB-1 

In this study, among beta-lactamase-positive beta-lactam-resistant *H. parainfluenzae*, type TEM beta-lactamase was the most dominant, found in 86.1% isolates; the ROB-1 gene was identified in 2.8%. Our findings are in general agreement with the first four years of the global PROTEKT study (duration 1999–2003) comprising 137 centers of 38 countries (including Poland) [22]. The overall prevalences of TEM-1 and ROB-1 positive *H. influenzae* isolates in this study were 93.7% and 4.6%, respectively. In Poland, authors found 95.0% TEM-1 positive isolates and none with ROB-1 beta-lactamase [22]. Similar results were obtained by Nakamura et al. [19] and Touati et al. [18], who did not isolate any *H. influenzae* clinical strain producing ROB-1 type beta-lactamase either from sputum or from nasopharynx, respectively. According to Garcia-Cobos et al. [21], of 40 clinical (e.g., from genital mucosae, urine, respiratory secretions, peritoneal fluid, blood) *H. parainfluenzae* isolates, 84.6% of *bla*^+^ isolates were identified as possessing TEM-1 beta-lactamases, and 7.7% possessed TEM-34 and TEM-182. According to the literature, strains positive for both TEM-1 and ROB-1 are rare [5,22], and conferring resistance to second generation cephalosporins [22] was also confirmed by this study. We also found that 30.6%, 19.4%, 16.7%, 11.1%, 8.3%, and 2.8% of *H. parainfluenzae* isolates from respiratory microbiota harbored the other following beta-lactamase genes: *bla*_OXA_, *bla*_DHA_, *bla*_SHV_, *bla*_GES_, *bla*_CMY_, and *bla*_VEB_, respectively. None of the tested isolates expressed *bla*_PER_ or *bla*_CTX-M-1_ genes.

Many difficulties with the cefinase test results have been reported, warning against its weak detection level of some beta-lactamases. There have been some conflicting reports about the sensitivity of nitrocefin hydrolysis, especially in detecting the ROB-1 beta-lactamase [5,22]. There have also been some cases where nitrocefin hydrolysis was positive and both TEM-1 or ROB-1 genes were negative in the PCR reaction, accounting for 1% of tested isolates, which was supposedly due to unrecognized beta-lactamase in that study [22]. According to Tinguely et al. [7], who reported in Switzerland a case of XDR *H. parainfluenzae* isolate which carried the *bla*_TEM-1_ gene, the phenotypically produced beta-lactamase was not expressed according to the nitrocefin test (for both the cefinase paper disc and the hydrolytic activity against nitrocefin).

### 4.6. Discrepancies in Phenotype and Genotype Classification for H. parainfluenzae 

Furthermore, many authors have pointed to the imperfection of the EUCAST recommendations and the diagnostic scheme of haemophilic bacteria in the context of detection and phenotypic activity of beta-lactamases [44,45]. It is often underlined that phenotypic methods (e.g., in vitro sensitivity to ampicillin, amoxicillin, or amoxicillin with clavulanic acid) could lead to faulty results. According to Garcia-Cobos et al. [21], phenotype and genotype classification for *H. parainfluenzae* are the same as for *H. influenzae*. For this reason, it has been suggested that the scheme given by EUCAST [32,42] does not lead to the extraction of *H. influenzae* and/or *H. parainfluenzae* strains belonging to particular phenotypes of resistance to beta-lactams, including BLNAS, BLNAR, BLPAR, or BLPACR [44].

In our study, among 65.5% beta-lactam-resistant *H. parainfluenzae* isolates, gBLPAR genotype was the most common, found in 19.3% of isolates, followed by gBLNAS, found in 17.5% of isolates, gBLNAR II and gBLPAS as well as gBLPACR I in 15.8% of isolates, and gBLPBS I in 8.8% of isolates. Genotypes low-gBLNAR I, low-gBLNAR II, gBLPBS II, and gBLPACR II were determined in one isolate each. Among beta-lactam-resistant isolates, 36.8% were *bla*^−^, classified into gBLNAS, low-gBLNAR, and gBLNAR genotypes. In Garcia-Cobos’s et al. [21] study, gBLNAS, gBLNAR, gBLPAR, and gBLPACR genotypes were detected, accounting for 50%, 17.5%, 20%, and 12.5%, respectively. In the same research, all beta-lactamase-positive isolates were ampicillin- and amoxicillin-resistant, among which 15.4% were gBLPAR and 7.7% were gBLPACR.

### 4.7. Polymorphism of the ftsI Resistance Gene 

At the same time, the need to analyze the polymorphisms of resistance genes (e.g., the *ftsI* gene) is strongly indicated. It also follows that not all changes in the amino acid structure of PBP3 proteins are related to the determined phenotype of in vitro resistance [44]. This fact has been particularly emphasized in the case of strains with the genotype gBLNAR, which phenotypically show sensitivity to ampicillin, amoxicillin, and amoxicillin with clavulanic acid [44]. Similar observations have been made concerning strains belonging to the BLNAS phenotype, which are often positive in the cefinase assay, but not found to contain beta-lactamase genes [45].

In our study, among all *H. parainfluenzae* isolates tested, 37.9% were *ftsI*^+^, among which 78.8% were beta-lactam-resistant. The most frequent amino acid substitution was at the Lys-526 position of the *ftsI* gene, found in 93.8% of *ftsI*^+^
*bla*^+^ isolates, whereas the Thr-385 substitution was found in 12.5%. Generally, in *H. parainfluenzae*, numerous amino acid substitutions in PBP3 have been reported as follows: Lys-276-Asn, Ala-307-Asn, Val-329-Ile, Ile-442-Phe, Val-511-Ala, Asn-526-Lys, Asn-526-Ser, Ala-343-Val, Asn-526-His, Ala-530-Ser, and Thr-574-Ala [7,19,21,23,46]. According to Hasegawa et al. [8], the substitutions Arg-517-His, Asp-526-Lys, and Ser-385-Thr in the *ftsI* gene affected the degree of resistance to beta-lactams. According to Garcia-Cobos et al. [21], beta-lactamase-positive *H. parainfluenzae* clinical isolates had Met-69-Val, Met-69-Ile, Trp-165-Ile, and/or Arg-275-Leu amino acid substitutions. Wienholtz et al. [13] also selected positions 385, 511, and 526 as the most sensitive for site-directed mutagenesis in the *H. parainfluenzae ftsI* gene, with Val-511-Ala, Ile-442-Phe, and Val-526-Leu/Ile substitutions as the most frequent. In a case of XDR *H. parainfluenzae* isolates from urethral swab, Tinguely et al. [7] detected extra Ala-307-Asn and Val-329-Ile PBP3 substitutions that had never been described in a single *H. parainfluenzae* isolate. Additionally, Asn-526-Ser, Asn-526-Lys, or Ser-385-Thr mutations of the *ftsI* gene have been reported as the most frequent in *H. parainfluenzae* isolates, in agreement with our results, with Asn-526-Ser as the most species-specific [21]. This fact has also been reported by others [13], where the Asn-526-Ser mutation in the *ftsI* gene has been observed in almost all *H. parainfluenzae* isolates resistant to beta-lactam antibiotics. It has also been reported that some amino acid substitutions detected in the *ftsI* gene (PBP3 protein) corresponded to a resistance phenotype, conferring resistance to one or more beta-lactam antibiotics [13,21,45]. For example, the Ser-385-Thr substitution in *H. influenzae* was closely related to higher cefotaxime and cefixime MIC values, while Val-511-Ala increased the amoxicillin and amoxicillin–clavulanate 2/1 µg/mL MICs of gBLNAR *H. influenzae* and *H. parainfluenzae* isolates, respectively [21]. Val-511-Ala conferred resistance to ampicillin only in combination with any of three substitutions Asn-526-His, Asn-526-Ser, or Asn-526-Lys in *H. parainfluenzae* strains. Moreover, Ser-385-Thr in the presence of Val-511-Ala doubled the MIC values of extended spectrum cephalosporins, and when substitutions at all three positions were observed, an approximate 9-fold increase of cefotaxime MIC values were measured [13].

In this study, 28.1% of all commensal *H. parainfluenzae* isolates were beta-lactam-resistant *ftsI*^+^
*bla*^+^, indicating the existence of two simultaneous mechanisms of beta-lactam resistance, defined as gBLPACR I/II or gBLPBS I/II. This type of isolates is still rare, and their detection is difficult [5,18]. Touati et al. [18] observed only 6% of these isolates among tested nasopharyngeal *H. influenzae* strains. The same authors indicated that for these strains, resistance to amoxicillin–clavulanate is doubtful.

### 4.8. Beta-Lactamase-Negative ftsI-Positive Isolates

Furthermore, we found that 52.4% of gBLNAR or low-gBLNAR *bla*^−^ isolates had an *ftsI* gene with a substitution at the position Lys-526. This was in agreement with Touati et al. [18], who found that *H. influenzae* nasopharyngeal isolates had the *bla*_TEM-1_ gene but did not phenotypically produce any beta-lactamases (one isolate), or were BLNAR *ftsI*^−^ and *bla*_TEM_-negative. Some researchers have suggested the possibility of horizontal transfer of genes [14], including the *ftsI* gene, among *H. influenzae* and *H. parainfluenzae* clinical isolates. This has been supported, for example, by a transfer of Asn-526-Lys amino acid substitution specific to *H. influenzae* and now also reported in *H. parainfluenzae* isolates [21]. This confirmed that *H. parainfluenzae* might be a huge reservoir of multiple beta-lactamase-carrying plasmids for other bacterial species [14]. This is an important issue because exact determination of the mechanism of resistance (production of beta-lactamases and/or mutations in the *ftsI* gene) to beta-lactam antibiotics in haemophilic bacteria will enable the implementation of effective therapeutic options to limit the spread of this phenomenon.

The authors are aware of the limitations associated with this study, especially related to the low number of samples. The authors also note that official breakpoints recommended by EUCAST were not available for *H. parainfluenzae*, only for *H. influenzae* in general, which may not be sufficient nowadays. Furthermore, mapping of *bla*_TEM_ gene, P3 promoter variants, and detection of other amino acid substitutions in the *ftsI* gene need to be investigated to analyze the degree of beta-lactamase expression among *H. parainfluenzae* isolates.

In conclusion, this was the first study to highlight the high diversity of beta-lactamase genes among commensal *H. parainfluenzae* isolates from respiratory microbiota in eastern Poland. This study showed the high sensitivity and specificity of the benzylpenicillin screen test, as well as PCR detection of *bla* genes for *H. parainfluenzae*. These methods may be a useful tool for the rapid screening of beta-lactamase prevalence among *H. parainfluenzae* clinical isolates resistant to beta-lactam antibiotics.

## Figures and Tables

**Figure 1 microorganisms-07-00427-f001:**
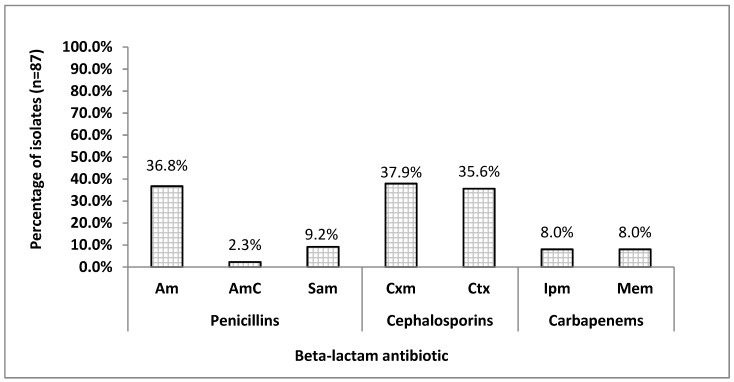
Distribution of resistance to beta-lactam antibiotics among *Haemophilus parainfluenzae* isolates (*n* = 87). Am—ampicillin, Cxm—cefuroxime (oral), Ctx—cefotaxime, Ipm—imipenem, Mem—meropenem, AmC—amoxicillin–clavulanate, Sam—ampicillin–sulbactam.

**Figure 2 microorganisms-07-00427-f002:**
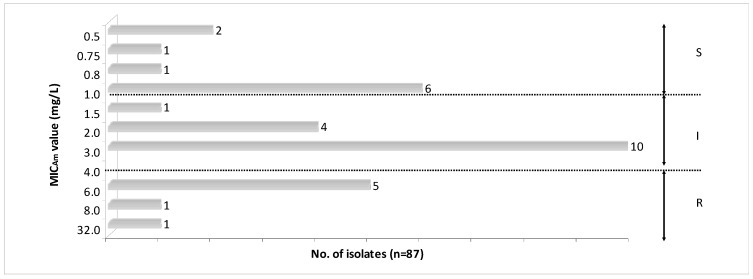
MIC_Am_ values for ampicillin among *Haemophilus parainfluenzae* isolates from respiratory microbiota. S—susceptible isolates, I—susceptible, increased exposure (formerly intermediate) isolates, R—resistant isolates.

**Figure 3 microorganisms-07-00427-f003:**
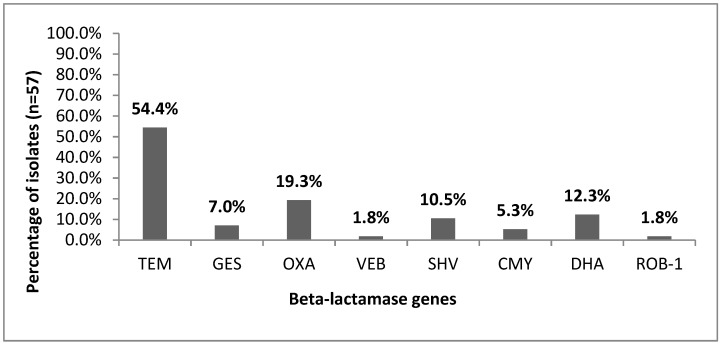
Presence of *bla* genes among *Haemophilus parainfluenzae* isolates (*n* = 57).

**Table 1 microorganisms-07-00427-t001:** Oligonucleotides used as primers for amplification.

bla Gene	Primer	Sequence (5′to 3′)	Product Size (bp)	Reference
*bla_TEM-1_*	*TEM-F*	ATTCTTGAAGACGAAAGGGC	1150	[35]
	*TEM-R*	ACGCTCAGTGGAACGAAAAC		
*bla_GES_*	*GESf*	TTCCATCTCAAGGGATCACC	890	[36]
	*GESr*	GCGTCAACTATTTGTCCGTG		
*bla_OXA_*	*OXA-F*	AGTGTGTTTAGAATGGTGATC	813	[37]
	*OXA-R*	GTTAGCGGTAATTTAACCAGATAG		
*bla_VEB_*	*VEB-F*	GTTAGCGGTAATTTAACCAGATAG	1070	[38]
	*VEB-R*	CGGTTTGGGCTATGGGCAG		
*bla_CTX-M-1_*	*P1C*	TTAATTCGTCTCTTCCAGA	1000	[27,39]
	*P2D*	CAGCGCTTTTGCCGTCTAAG		
*bla_SHV_*	*SHV-A*	ACTGAATGAGGCGCTTCC	300	[40]
	*SHV-B*	ATCCCGCAGATAAATCACC		
*bla_CMY_*	*CMY-F*	CAATGTGTGAGAAGCAGTC	1432	[26]
	*CMY-R*	CGCATGGGATTTTCCTTGCTG		
*bla_DHA_*	*DHA-f*	AACTTTCACAGGTGTGCTGGGT	405	[28]
	*DHA-r*	CCGTACGCATACTGGCTTTGC		
*bla_PER_*	*PER-F*	TGACGATCTGGAACCTTT	900	[41]
	*PER-R*	AACTGCATAACCTACTCC		
*bla_ROB-1_*	*ROB-f*	GGATCAGAGTAATAATTTCTG	192	[17]
	*ROB-r*	GCCATTGAAAGCAAGTTTCAACGG		
*pbp3-BLN*	*BLN-F*	GTCACACCACGGTTACTTGAA	465	[19]
	*BLN-R*	CCCGCAGTAAATGCCACATATTTC		
*pbp3-INT*	*INT-F*	GATACTACGTCCTTTAAATTAAGCG	554	[19]
	*INT-R*	CCCGCAGTAAATGCCACATATTTC		

**Table 2 microorganisms-07-00427-t002:** Classification of phenotypes of *Haemophilus parainfluenzae* isolates based on the results of drug susceptibility to beta-lactams.

**Phenotype**	**Description**
BLNAS	beta-lactamase-negative cefinase-negative ampicillin-susceptible isolate
BLNAI	beta-lactamase negative cefinase-negative isolate with reduced susceptibility to ampicillin
low-BLNAR	low-level BLNAR isolate; beta-lactamase-negative ampicillin-resistant isolate with ampicillin MICs in the range of 0.5–2.0 mg/L
BLNAR	beta-lactamase-negative ampicillin-resistant isolate with ampicillin MICs ≥ 2.0 mg/L
BLNBR	beta-lactamase negative cefinase-negative isolate resistant to one or more beta-lactams (benzylpenicillin, ampicillin, cephalosporins, or carbapenems)
BLPAS	beta-lactamase-positive cefinase-negative ampicillin-susceptible isolate
BLPAI	beta-lactamase-positive cefinase-negative isolate with reduced susceptibility to ampicillin
BLPAR	beta-lactamase-positive cefinase-positive ampicillin- and benzylpenicillin-resistant amoxicillin-clavulanic acid-susceptible isolate
BLPACR	beta-lactamase-positive cefinase-negative ampicillin-clavulanic acid-, ampicillin-, or benzylpenicillin-resistant isolate
**Genotype**	**Description**
gBLNAS	isolate negative for beta-lactamase genes ampicillin-susceptible without any amino acid substitutions in *ftsI* gene
gBLNAR	beta-lactamase-negative ampicillin-resistant isolate positive for β-lactamase genes with *ftsI* gene mutations: subgroup I—substitution of Arg-517→His-517 (Arg-517-His); II—substitution of Arg-526→Lys-526 (Arg-526-Lys); IIa—substitution at the position of 526 except Ala-502; IIb—substitution of Val-502→Ala-502 (Val-502-Ala); IIc—substitution of Thr-502→Ala-502 (Thr-502-Ala); IId—substitution of Val-449→Ile-449 (Val-449-Ile); III—substitutions of three amino acids Met-377→Ile-377, Ser-385→Thr-385 and Leu-389→Phe-389 with addition of Asn-526-Lys
low-gBLNAR	low-level gBLNAR beta-lactamase-negative isolate negative for β-lactamase genes isolate: subgroup I—without amino acid substitution; II—substitution at the Lys-526 position in *ftsI* gene
gBLPAR	isolate positive for beta-lactamase genes ampicillin-resistant without any amino acid substitutions in *ftsI* gene
gBLPACR	isolate positive for beta-lactamase genes ampicillin-resistant with *ftsI* gene mutations: subgroup I—substitutions of Arg-517-His and Arg-526-Lys; II—substitutions of Met-377-Ile, Ser-385-Thr, Leu-389-Phe, and Asn-526-Lys
gBLPBS	isolate positive for beta-lactamase genes with *ftsI* gene mutations: subgroup I—substitution at the Lys-526 position; II—substitutions at the Thr-385 and Lys-526 positions

**Table 3 microorganisms-07-00427-t003:** Statistical diagnostic values of conventional phenotypic methods used to determine beta-lactamase production among *Haemophilus parainfluenzae* isolates in comparison to PCR amplification results.

Phenotypical Method	Sensitivity (%)	Specificity (%)	PPV ^1^ (%)	NPV ^2^ (%)
cefinase test	52.25	100.0	100.0	34.57
penicillinase test (API NH)	54.72	96.43	90.91	36.84
amoxicillin–clavulanate	51.33	100.0	100.0	33.73
benzylpenicillin screen	90.63	100.0	100.0	82.35

^1^ PPV—Positive predictive value; ^2^ NPV—Negative predictive value.

**Table 4 microorganisms-07-00427-t004:** Distribution of resistance genes of the 57 *Haemophilus parainfluenzae* isolates and resistance patterns to seven beta-lactam antibiotics.

No.	Isolate Name	cef ^1^	Resistance Pattern	MIC_Am_ ^9^ (mg/L)	Beta-Lactamase Gene							*ftsI* ^10^	*ftsI* Substitution		Phenotype	Genotype
					TEM-1	GES	OXA	VEB	SHV	CMY	DHA		Thr-385 ^11^/Lys-526 ^12^	Lys-526		
beta-lactamase-positive isolates																
1.	2AU	-	Cxm ^3^ Ctx ^4^	-	x		x								BLPAS	gBLPAS
2.	2BU	-	Am ^2^ Cxm Ctx	3.0	x							+		+	BLPACR	gBLPACR I
3.	2CU	-	Am Sam ^8^ Cxm	2.0	x							+		+	BLPACR	gBLPACR I
4.	5BU	-	Am Sam	3.0	x										BLPACR	gBLPAR
5.	11BU	-	Cxm Ctx	-	x							+		+	BLPAS	gBLPBS I
6.	23BU	-	Am Cxm Ctx	3.0	x							+		+	BLPACR	gBLPACR I
7.	24AU	-	Am Cxm Ctx	2.0	x						x	+		+	BLPACR	gBLPACR I
8.	27CU	-	Ctx	-	x							+		+	BLPAS	gBLPBS I
9.	28BU	-	Ctx	-	x							+	+	+	BLPAS	gBLPBS II
10.	28CU	-	Ipm ^5^	-	x							+		+	BLPAS	gBLPBS I
11.	50AU	-	Cxm	-	x							+		+	BLPAS	gBLPBS I
12.	50CU	-	Am	1.0	x										BLPAR	gBLPAR
13.	W1HB	-	Am	1.0	x		x		x		x				BLPAR	gBLPAR
14.	W1HE	-	Am	1.5	x										BLPAR	gBLPAR
15.	W4HB	-	Cxm Ctx Ipm Mem ^6^	-							x	+		+	BLPAS	gBLPBS I
16.	W4HC	+	Am AmC ^7^ Cxm Ctx Ipm Mem	32.0					x			+		+	BLPBR	gBLPACR I
17.	W5HD	+	Am Cxm	6.0	x							+		+	BLPAR	gBLPACR I
18.	W5HP	+	Am AmC Cxm Ctx	1.0	x						x	+		+	BLPBR	gBLPACR I
19.	W6HB	-	Am Cxm	6.0							x				BLPAR	gBLPAR
20.	W7HC	-	Am Mem	1.0	x							+		+	BLPACR	gBLPACR I
21.	W12HB	-	Am Cxm Ctx	1.0	x										BLPAR	gBLPAR
22.	IM 1GB	-	Am Cxm Ctx Ipm Mem	0.5	x	x	x		x	x		+		+	BLPACR	gBLPACR I
23.	IM 2GB	-	Cxm	-	x										BLPAS	gBLPAS
24.	IM 4GB	+	Am	0.75	x	x	x								BLPAR	gBLPAR
25.	IM 5GB	-	Cxm	-	x										BLPAS	gBLPAS
26.	IM 5GC	-	Am Cxm Ctx	0.5	x							+	+		BLPACR	gBLPACR II
27.	IM 6GB	-	Cxm Ctx Mem	-	x										BLPAS	gBLPAS
28.	IM 6NLB	-	Cxm Ipm	-	x		x								BLPAS	gBLPAS
29.	IM 9GB	-	Am	6.0	x										BLPAR	gBLPAR
30.	IM 9GE	-	Cxm	-	x	x	x								BLPAS	gBLPAS
31.	IM 10GB	-	Am	6.0	x			x	x						BLPAR	gBLPAR
32.	IM 12NC	-	Cxm	-	x		x								BLPAS	gBLPAS
33.	IM 12GB	-	Cxm Ctx	-	x										BLPAS	gBLPAS
34.	IM 14GC	-	Ctx	-	x	x				x					BLPAS	gBLPAS
35.	IM 18GA	-	Am	3.0							x				BLPAR	gBLPAR
36.	IM 20GB	-	Am	1.0					x						BLPAR	gBLPAR
beta-lactamase-negative *ftsI*-positive isolates																
1.	10BU	-	Am Sam Cxm Ctx	2.0								+		+	BLNAR	gBLNAR II
2.	11AU	-	Am Cxm	3.0								+		+	BLNAR	gBLNAR II
3.	23CU	-	Am Cxm Ctx	3.0								+		+	BLNAR	gBLNAR II
4.	24GU	-	Am Cxm Ctx	6.0								+		+	BLNAR	gBLNAR II
5.	25BU	-	Am Cxm Ctx	8.0								+		+	BLNAR	gBLNAR II
6.	39CU	+	Am Cxm Ctx	3.0								+		+	BLPAR	gBLNAR II
7.	W1HC	-	Am	0.8								+		+	low-BLNAR	low-gBLNAR II
8.	W2HA	-	Am Cxm Ctx	3.0								+		+	BLNAR	gBLNAR II
9.	W3HA	-	Am Cxm Ctx	3.0								+		+	BLNAR	gBLNAR II
10.	W3HB	-	Am Cxm Ctx	3.0								+		+	BLNAR	gBLNAR II
11.	IM 18GB	-	Am	2.0											low-BLNAR	low-gBLNAR I
beta-lactamase-negative *ftsI*-negative isolates																
1.	4AU	-	Ipm	-											BLNAS	gBLNAS
2.	6BU	-	Cxm Ctx	-											BLNAS	gBLNAS
3.	7AU	-	Ctx	-											BLNAS	gBLNAS
4.	10AU	-	Ipm	-											BLNAS	gBLNAS
5.	22AU	-	Ctx Mem	-											BLNAS	gBLNAS
6.	25CU	-	Cxm Mem	-											BLNAS	gBLNAS
7.	26CU	-	Ctx	-											BLNAS	gBLNAS
8.	27BU	-	Ctx	-											BLNAS	gBLNAS
9.	43AU	-	Ctx	-											BLNAS	gBLNAS
10.	47BU	-	Ctx	-											BLNAS	gBLNAS

^1^ cef—cefinase test, ^2^ Am—ampicillin, ^3^ Cxm—cefuroxime (oral), ^4^ Ctx—cefotaxime, ^5^ Ipm—imipenem, ^6^ Mem—meropenem, ^7^ AmC—amoxicillin–clavulanate, ^8^ Sam—ampicillin–sulbactam, ^9^ MIC_Am_—minimal inhibitory concentration values for ampicillin, ^10^
*ftsI*—gene encoding the transpeptidase domain of PBP3 protein, ^11^ Thr-385 − Ser-385-Thr amino acid substitution in *ftsI* gene, ^12^ Lys-526 − Asp-526-Lys amino acid substitution in *ftsI* gene.

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
