# Peer review of "Prevalence of Resistance to β-Lactam Antibiotics and bla Genes Among Commensal Haemophilus parainfluenzae Isolates from Respiratory Microbiota in Poland"

_microorganisms, 2019, doi:10.3390/microorganisms7100427_

Round 1
Reviewer 1:
1.The study by S. Andrzejczuk et.al analyzed resistance to beta-lactam antibiotics in a human pathogen Haemophilus parainfluenzae. The authors screened about 90 clinical isolates and assayed for the phenotypic and genotypic determinants of the resistance. The data show a large diversity of the bla genes encoded by the isolates indicating a furthermore growing concern about the resistance of this pathogen. Overall, the data are of significance to the field. Several concerns are outlined below.
Answer 1: Specifically, reflecting the opinion of the Reviewer, additional revised parts and information were added to the manuscript. Based on comments from the Reviewer, the text was partially modified.
All other concerns raised by Reviewer are carefully addressed in the point by point responses below and changes made in this revised manuscript appear in red font. This manuscript has been substantially revised based on the helpful input of the reviewer, including updated data, figures, and revised discussion. The reviewers’ recommendations improved the quality of this manuscript and we hope that You will find it suitable for publication in Microorganisms
2.The authors targeted pbp3 gene for mutation analysis, without explaining whether other homologs of this gene are commonly present in the genomes of these organisms. A more serious concern is about detecting mutation in the gene by PCR only, without sequencing to prove. How reliable is it? Do the authors have any controls verifying their PCR results? What about other potential mutations within the gene?
Answer 2: It has been mentioned in the Introduction section (Lines 53-54) by the sentence: ‘In H. influenzae five different PBPs were frequently reported, and PBP3 as the most correlated with resistance to beta-lactams.’
We agree with the Reviewer that sequencing of the amplification product would be better to improve the mutation. However, detecting mutation by sequencing is quite expensive for us, especially for our 33 strains. This study was a part of doctoral dissertation. That were a preliminary studies conducted on a large number of opportunistic fastidious bacterial isolates with increased requirements. Initially, we did not plan sequencing method in the budget of this work. So we had could not do sequencing. That is why we made mutation detection using a cheaper method developed by Nakamura et al. (2009) to determine of the likely mutations (in our opinion it is more interesting then detection of the pbp3 gene alone) and which could be used in routine diagnostic activities - cheaper than sequencing. Nevertheless, these two amino acids point mutations were undoubtedly the most frequently observed in H. parainfluenzae, what was supported by many reports cited in the text. In addition, we are going to apply for a grant that would allow us to carry out more accurate genetic tests what is connected with obtained result
3.The authors need to explain clearly what methods were used and for what reason. L. 182-183 are confusing. It is also not clear how “uncertain” result is defined and what is the significance of such result (L.185).
200 says 54.4% were TEM-1-positive. However, only TEM is shown in the figure and later mentioned in the text, besides, the figure shows 86.1% for TEM.
Answer 3: We apologize for unclear wording, which confused and raised the Reviewer's doubts. In fact, this phrase needs to be explained what we have done in Line 212 by addition of ‘(PEN in API NH strip) gave not clear, uncertain or ambiguous result (e.g. a weak reaction)’ phrase. The results of phenotypic studies based on biochemical and morphological traits were sometimes less precise and difficult to interpret compared to data obtained, e.g., using the molecular method, and do not allow for unambiguous determination of the H. parainfluenzae species. Both reports, our own results as well as the experience of many researchers confirm the existence of limitations in the use of phenotypic methods and point to a number of problems related to the identification of microorganisms, especially fastidious ones with increased requirements. These limitations and the assessment of phenotypic features of microorganisms were already reported by other authors (e.g. Kosikowska et al.,2015, entitled: ‘Application of MALDI-TOF MS for identification of clinical isolates of bacteria from humans and animals’).
Percentages shown in figure have been corrected and the name of TEM-beta-lactamase has been changed to TEM-1 beta-lactamase.
4.All the abbreviations need to be spelled out in the text (not only in the legend). Since there are so many of them, it would help to categorize them, which could be done in a table format. Table 2 and its legend are too difficult to follow due to all these abbreviations. Besides, some naming is confusing, for example, how gBLPAR could be a genotype, if it means “amp resistant”, which is a phenotype.
Answer 4: We admit that the Reviewer was right that these shortcuts were burdensome for the reader and we have explained everything in added Table 2 in the Materials in methods section.
Table 2. Classification of phenotypes of H. parainfluenzae isolates based on the results of drug susceptibility to beta-lactams.
|
Phenotype |
Description |
|
BLNAS |
beta-lactamase-negative cefinase-negative ampicillin-susceptible isolate |
|
BLNAI |
beta-lactamase negative cefinase-negative isolate with reduced susceptibility to ampicillin |
|
low-BLNAR |
low-level BLNAR isolate; beta-lactamase-negative ampicillin-resistant isolate with ampicillin MICs in the range of 0.5 - 2.0 mg/L |
|
BLNAR |
beta-lactamase-negative ampicillin-resistant isolate with ampicillin MICs ≥ 2.0 mg/L |
|
BLNBR |
beta-lactamase negative cefinase-negative isolate resistant to one or more beta-lactams (benzylpenicillin, ampicillin, cephalosporins or carbapenems) |
|
BLPAS |
beta-lactamase-positive cefinase-negative ampicillin-susceptible isolate |
|
BLPAI |
beta-lactamase-positive cefinase-negative isolate with reduced susceptibility to ampicillin |
|
BLPAR |
beta-lactamase-positive cefinase-positive ampicillin- and benzylpenicillin-resistant amoxicillin-clavulanic acid-susceptible isolate |
|
BLPACR |
beta-lactamase-positive cefinase-negative ampicillin-clavulanic acid-, ampicillin- or benzylpenicillin-resistant isolate |
|
Genotype |
Description |
|
gBLNAS |
isolate negative for beta-lactamase genes ampicillin-susceptible without any amino-acid substitutions in ftsI gene |
|
gBLNAR |
beta-lactamase-negative ampicillin resistant isolate positive for β-lactamase genes with ftsI gene mutations: subgroup I - substitution of Arg-517→His-517 (Arg-517-His); II - substitution of Arg-526→Lys-526 (Arg-526-Lys); IIa - substitution at the position of 526 except Ala-502; IIb substitution of Val-502→Ala-502 (Val-502-Ala); IIc - substitution of Thr-502→Ala-502 (Thr-502-Ala); IId - substitution of Val-449→Ile-449 (Val-449-Ile); III - substitutions of three amino acids Met-377→Ile-377, Ser-385→Thr-385 and Leu-389→Phe-389 with addition of Asn-526-Lys |
|
low-gBLNAR |
low-level gBLNAR beta-lactamase-negative isolate negative for β-lactamase genes isolate: subgroup I - without amino-acid substitution; II - substitution at the Lys-526 position in ftsI gene |
|
gBLPAR |
isolate positive for beta-lactamase genes ampicillin-resistant without any amino-acid substitutions in ftsI gene |
|
gBLPACR |
isolate positive for beta-lactamase genes ampicillin-resistant with ftsI gene mutations: subgroup I - substitutions of Arg-517-His and Arg-526-Lys; II - substitutions of Met-377-Ile, Ser-385-Thr, Leu-389-Phe and Asn-526-Lys |
|
gBLPBS |
isolate positive for beta-lactamase genes with ftsI gene mutations: subgroup I - substitution at the Lys-526 position; II - substitutions at the Thr-385 and Lys-526 positions |
The Legend of Table 3 has been limited to: ‘1 cef - cefinase test, 2 Am- ampicillin, 3 Cxm - cefuroxime (oral), 4 Ctx - cefotaxime, 5 Ipm - imipenem, 6 Mem - meropenem, 7 AmC - amoxicillin-clavulanate, 8 Sam - ampicillin-sulbactam, 9 MICAm - minimal inhibitory concentration values for ampicillin, 10 ftsI - gene encoding the transpeptidase domain of PBP3 protein, 11 Thr-385 - Ser-385-Thr amino-acid substitution in ftsI gene, 12 Lys-526 - Asp-526-Lys amino-acid substitution in ftsI gene’
5.In material and methods, sensitivity, specificity, positive and negative predictive values need to be briefly defined and their calculation briefly explained.
Answer 5: It has been explained by addition of paragraph: ‘Sensitivity defined as the percentage of true positives was calculated as the proortion of H. parainfluenzae isolates with positive test among all isolates tested which actually produced beta-lactamase, whereas specificity defined as the percentage of true negatives was calculated as the proportion of isolates which test negative among all isolates which actually do not produce beta-lactamase. PPV and NPV allowed to clinically obtain among H. parainfluenzae isolates tested how likely a production of beta-lactamase was in comparison to PCR results. Positive predictive was the probability that following a positive test result, that individual bacterial isolate will truly produce that specific beta-lactamase, while negative predictive value was the probability that following a negative test result, that individual bacterial isolate will truly not produce that specific enzyme.
6.Fig.1 and 3 show similar type of data, but using number of isolates or percentage. It would help to present such data using consistent format. Fig.1 does not show the insert mentioned in the legend. Does it need to be in color? Are any of these isolates resistant to multiple antibiotics? The legend says “ability to synthesize beta-lactamase”, which is not shown.
Answer 6: We thank the Reviewer for our shortcomings and for valuable tips. Although Fig.1 and 3 do not show similar type of data - resistance to a particular beta-lactam is one thing and the genes are another matter. Figure 1 shows the results of phenotypic drug susceptibility to beta-lactam antibiotics, while Figure 3 shows the occurrence of bla genes obtained. With reference to the Reviewer's comments, small chart (insert) - an internal graph showing how many strains produced beta-lactamases based on 3 phenotypic methods tested, how many "uncertain" and how negative results we obtained, has been removed from Fig. 1. That small chart probably has not been loaded in the version sent to reviewers. We agree with the Reviewer that it will be much more easier to analyze the results if data will consistently use a percentage format. The way the axes are signed has been standardized as far as possible.
Figure 1. Distribution of resistance to beta-lactam antibiotics among Haemophilus parainfluenzae isolates (n=87).
Furthermore, as defined by a group of international experts of the European Centre for Disease Prevention and Control (ECDC) and the Centers for Disease Control and Prevention (CDC), MDR strains are resistant to at least one antimicrobial agent in three or more antimicrobial classes. Therefore, because we are talking about beta-lactams, these isolates were not characterized as multi-drug resistant. However, the text contains information about the number of MDR isolates resistant to more than one antibiotic from the studied group of drugs.
7.The correlations between the data obtained by different assays are potentially insightful, but need to be explained better. The language like “benzylpenicillin disk screen test is used to exclude beta-lactamase resistance mechanisms” (L.309) needs to be clarified. The following sentences are not clear either (L. 310-312).
Answer 7: We are aware of strong discrepancies with the interpretation of the results, depending on which test we used, and which may affect the effectiveness of any treatment. example if in the API test, the PEN test is positive, penicillins cannot be used and then additional drug sensitivity testes for beta-lactams should be performed what the manufacturer explained. The use of penicillins (e.g. penicillin G, amino-, carboxy- and ureidopenicillins is prohibited. However, according to EUCAST guidelines only a 1 U benzylpenicillin disc is used, and depending on the test result - positive or negative, you cannot or you can use beta-lactams. More precisely, according to the latest EUCAST recommendations, the flow chart of results interpretation for the benzylpenicillin 1 unit disk screen test is available. The test shall be used to exclude beta-lactam resistance mechanisms. When the screen is negative which means inhibition zone ≥12 mm all beta-lactam agents for which clinical breakpoints are available, including those with “Note”, can be reported susceptible without further testing. When the screen is positive which means inhibition zone <12 mm, possibility of beta-lactamase production and/or PBP3 mutations should be taken into account. As for ampicillin, amoxicillin and piperacillin (without beta-lactamase inhibitor), if beta-lactamase is positive, we should report isolate as resistant, and if beta-lactamase is negative we report susceptibility according to the clinical breakpoints for the agent in question. As for other beta-lactam agents except cefepime, cefpodoxime and imipenem we report susceptibility according to the clinical breakpoints for the agent in question. Additionally, for cefepime, cefpodoxime and imipenem, if resistant by both screen and agent disk diffusion test, we report isolate as resistant. If isolate is resistant by screen test and susceptible by agent disk diffusion test, we should determine the MIC of the agent and interpret according to breakpoints. We have added a more detailed explanation in the Materials and methods section (Lines 113-124), as well as in the Results (lines 326-332): ‘When the screen is negative all beta-lactam agents, for which clinical breakpoints are available, can be reported sensitive without further testing. This result excludes both beta-lactamase production and other beta-lactam resistance mechanisms. All positive screen results indicate a possibility of both resistance mechanisms - beta-lactamase production and/or PBP3 mutations as it was explained in the Materials and methods section based on EUCAST flow chart [44]. At the same time, additional results should be taken into account, e.g. ampicillin and amoxicillin susceptibility.’
8. Overall, a number of sentences require editing for clarity, terminology, and/or English. Abstract is difficult to follow, particularly, point 3. Discussion needs to be restructured, and the main points of discussion need to be emphasized, preferably with the first sentence in each paragraph clearly leading towards its content.
Answer 8: It has been corrected
Abstract is difficult to follow, particularly, point 3.
Abstract has been re-write ‘Results: Among 57 beta-lactam-resistant isolates, 63.2% encoded bla genes: blaTEM-1 predominated 54.4%, followed by blaOXA 19.3%, blaDHA 12.3%, blaSHV 10.5%, blaGES 7.0%, blaCMY 5.3%, blaVEB 1.8% and blaROB-1 1.8%. Lys-526 was the most common substitution in ftsI gene. The resistance genotypes were as follows: gBLNAS 17.5%, low-gBLNAR I 1.8%, low-gBLNAR II 1.8%, gBLNAR II 15.8%, gBLPAS 15.8%, gBLPAR 19.3%, gBLPBS I 8.8% and gBLPBS II 1.8%’
Discussion needs to be restructured, and the main points of discussion need to be emphasized, preferably with the first sentence in each paragraph clearly leading towards its content.
It has been done by addition of bolded phrases: ‘Resistance to beta-lactams among H. parainfluenzae’ (line 282); ‘The susceptibility to beta-lactam antibiotics among H. parainfluenzae isolates.’ (line 296); ‘Conventional methods for detection of beta-lactamase in haemophili.’ (line319); ‘PCR amplification as rapid detection method of beta-lactamases production.’ (line 341); ‘Prevalence of beta-lactamase type TEM-1 and ROB-1.’ (line 352); ‘Discrepancies in phenotype and genotype classification for H. parainfluenzae.’ (line 378); ‘Polymorphism of the ftsI resistance gene.’ (line 396); ‘Beta-lactamase-negative ftsI-positive isolates.’ (line 436).
Other comments
9. L. 20 “the” instead of “that” The sentence needs editing. L. 55 Could cite a recent review(s) instead of the large number of references. L.56 Consider “production of beta-lactamases” instead L. This is not in-vitro mechanisms, but an in-vitro approach? L. 90, 93 Consider “frozen stock” instead of “bacterial suspension” L.138-139 The genes are listed here, but throughout the text they are called without “bla”. It would help to explain it here. L. 140 Rephrase for clarity Include the primers PBP3-BLN and PBP3-INT into the table 1 Fig.2 legend: what is the reference for? L. 202 not “expressed”, but “encoded”. Use italic for all the gene and strain names. L. 260 Need to explain why the isolates are first called resistant, and then some of them are called sensitive. L. 292 Are these numbers of isolates or percent?
Answer 9: L.59 Consider “production of beta-lactamases” instead L. This is not in-vitro mechanisms, but an in-vitro approach? - phrase was re-write
94, 97 Consider “frozen stock” instead of “bacterial suspension” - it has been changed 149-151 The genes are listed here, but throughout the text they are called without “bla”. It would help to explain it here. - the phrase in brackets has been added ‘(called also further: TEM, GES, OXA, VEB, CTX-M-1, SHV, CMY, DHA, PER, ROB-1, respectively)’ 140 (now lines 151-153) Rephrase for clarity - sentence was re-write ‘PCR amplification was performed of the part of abovementioned genes from genomic DNA. Table 1 presents previously published sequences of oligonucleotides’
Include the primers PBP3-BLN and PBP3-INT into the table 1 - it has been added
Table 1. Oligonucleotides used as primers for amplification.
|
bla gene |
Primer |
Sequence (5’to 3’) |
Product |
Reference |
|
blaTEM-1 |
TEM-F |
ATTCTTGAAGACGAAAGGGC |
1150 |
[35] |
|
TEM-R |
ACGCTCAGTGGAACGAAAAC |
|||
|
blaGES |
GESf |
TTCCATCTCAAGGGATCACC |
890 |
[36] |
|
GESr |
GCGTCAACTATTTGTCCGTG |
|||
|
blaOXA |
OXA-F |
AGTGTGTTTAGAATGGTGATC |
813 |
[37] |
|
OXA-R |
GTTAGCGGTAATTTAACCAGATAG |
|||
|
blaVEB |
VEB-F |
GTTAGCGGTAATTTAACCAGATAG |
1070 |
[38] |
|
VEB-R |
CGGTTTGGGCTATGGGCAG |
|||
|
blaCTX-M-1 |
P1C |
TTAATTCGTCTCTTCCAGA |
1000 |
[27,39] |
|
P2D |
CAGCGCTTTTGCCGTCTAAG |
|||
|
blaSHV |
SHV-A |
ACTGAATGAGGCGCTTCC |
300 |
[40] |
|
SHV-B |
ATCCCGCAGATAAATCACC |
|||
|
blaCMY |
CMY-F |
CAATGTGTGAGAAGCAGTC |
1432 |
[26] |
|
CMY-R |
CGCATGGGATTTTCCTTGCTG |
|||
|
blaDHA |
DHA-f |
AACTTTCACAGGTGTGCTGGGT |
405 |
[28] |
|
DHA-r |
CCGTACGCATACTGGCTTTGC |
|||
|
blaPER |
PER-F |
TGACGATCTGGAACCTTT |
900 |
[41] |
|
PER-R |
AACTGCATAACCTACTCC |
|||
|
blaROB-1 |
ROB-f |
GGATCAGAGTAATAATTTCTG |
192 |
[17] |
|
ROB-r |
GCCATTGAAAGCAAGTTTCAACGG |
|||
|
pbp3-BLN |
BLN-F |
GTCACACCACGGTTACTTGAA |
465 |
[19] |
|
BLN-R |
CCCGCAGTAAATGCCACATATTTC |
|||
|
pbp3-INT |
INT-F |
GATACTACGTCCTTTAAATTAAGCG |
554 |
[19] |
|
INT-R |
CCCGCAGTAAATGCCACATATTTC |
Fig.2 legend: what is the reference for? - the reference has been deleted
202 (now line 229) not “expressed”, but “encoded”. - it has been changed
Use italic for all the gene and strain names. - it has been checked and corrected
275-278 Need to explain why the isolates are first called resistant, and then some of them are called sensitive.
The misunderstanding arises probably from the selection of tests and recommendations. First values were based on drug susceptibility tests which determined each phenotype in isolates tested, while the second value was based on the E-test-derived MICs values which referred to EUCAST recommendations and breakpoints. Nevertheless, the problematic sentence has been corrected: ‘All BLPAR isolates were ampicillin resistant, one had MICAm = 0.75 mg/L (sensitive according to EUCAST), six had MICAm values ranged from 1.0 to 3.0 mg/L (intermediate according to EUCAST) and four had MICAm = 6.0 mg/L (resistant according to EUCAST).’
292 Are these numbers of isolates or percent? – The sentence has been deleted as redundant
Reviewer 2 Report
The idea is interesting.
Answer 1: Specifically, reflecting the opinion of the Reviewer, additional revised parts and information were added to the manuscript. Based on comments from the Reviewer, the text was partially modified.
All other concerns raised by Reviewer are carefully addressed in the point by point responses below and changes made in this revised manuscript appear in red font. This manuscript has been substantially revised based on the helpful input of the reviewer, including updated data, figures, and revised discussion. The reviewers’ recommendations improved the quality of this manuscript and we hope that You will find it suitable for publication in Microorganisms.
2. English language needs amelioration in several points, I have already made corrections but you should check it again.
Answer 2: English language needs amelioration in several points - the language of manuscript has been corrected
3. I suggest you should check again the discussion and especially the results.
Answer 3: Line 19 ‘and’ was added. The sentence goes: ‘Beta-lactams as the most frequently used antimicrobials, and are the first-line drugs in many infectious diseases, e.g. pneumonia, otitis media.’
Line 20 a comma after ‘Due to this fact….’ was added
Line 24 ‘the’ was added. The sentence goes: ‘Among 65.5% of beta-lactam-resistant isolates, 63.1% harbour the following bla genes…..’
Line 31 ‘useful tools’ were corrected into ‘these methods may be useful tool for’
Line 36 sentence was re-write: ‘Beta-lactam antibiotics and especially penicillins are most frequently used group of antimicrobials in the European Union.’
Line 39 ‘of’ was added in phrase ‘range of 5.0 - 7.2 DDD’
Line 44 ‘a’ was replaced by ‘for’ in phrase ‘Amoxicillin is recommended for the treatment….’
Line 49 ‘a’ was replaced by ‘the’ in phrase ‘Due to the very frequent and widespread use….’
Line 50 the sentence was re-write: ‘Due to the very frequent and widespread use of beta-lactams, the development of resistance to this group of drugs has been observed, also among isolates of H. influenzae and Haemophilus parainfluenzae, for many years [5–23].’
Lines 72-74 the sentence was re-write: ‘According to literature, the third strategy is possible consisting of both mechanisms of resistance to beta-lactam antibiotics are present in one strain of haemophilic bacteria, so that this strain can simultaneously produce beta-lactamases and has modifications in the structure of PBP proteins.’
Line 78 ‘a’ was deleted in phrase ‘detect ten different bla genes’
Line 182 phrase ‘insert shows a summarized result of beta-lactamase synthesis’ was deleted
Line 216 phrase was corrected ‘results (bla genes identification) are shown in Table 2’
Line 219 phrase was corrected ‘with a value of 96.43%’
Line 266 name ‘cefotaxime’ was corrected
Line 285 phrase was re-write ‘impact on therapeutic problems for infections’
Line 286 phrase was re-write ‘importance of this issue is among others evidenced by an increasing’
Lines 288-289 sentence was re-write ‘Since 2013 in PubMed database only a few publications exist about H. parainfluenzae resistance to beta-lactam antibiotics and their molecular mechanisms.’
Line 289 ‘however’ was replaced by ‘nevertheless’
Line 333 ‘this study’ was replaced by ‘our study’
Line 335 ‘what was’ was replaced by ‘which was’
Line 341 ‘undermines’ was replaced by ‘question’
Line 343 phrase was re-write ‘indicate at the same time treatment failure’
Lines 357-360 sentence was re-write ‘Similar results were obtained by Nakamura et al. [19] and Touati et al. [18], who did not isolate any H. influenzae clinical strain producing ROB-1 type beta-lactamase neither from sputum nor from nasopharynx, respectively’
Line 364 ‘and conferred resistance’ was replaced by ‘and conferring resistance’
Lines 365-367 sentence was re-write ‘We have also found that 30.6%, 19.4%, 16.7%, 11.1%, 8.3% and 2.8% of H. parainfluenzae isolates from respiratory microbiota harboured other following beta-lactamase genes: OXA, DHA, SHV, GES, CMY and VEB, respectively.’
Line 369 phrase ‘conflicting information’s’ was replaced by ‘conflicting reports’
Line 370 phrase ‘that nitrocefin hydrolysis was positive’ was replaced by ‘where nitrocefin hydrolysis was positive’
Line 372 ‘for’ was added in a phrase ‘accounting for 1%’
Lines 373-376 sentence was re-write ‘According to Tinguely et al. [7], who reported in Switzerland a case of XDR H. parainfluenzae isolate which carried blaTEM-1 gene, the phenotypically produced beta-lactamase was not expressed on the basis of nitrocefin test (both the cefinase paper disc and the hydrolytic activity against nitrocefin).’
Line 402 word ‘tested’ changed placed in a phrase ‘among all H. parainfluenzae isolates tested’
Line 403 phrase was re-write ‘The most frequent amino-acid substitution was….’
Line 408 ‘on’ was deleted from phrase ‘affected the degree’
Line 414 ‘an’ was deleted from phrase ‘detected extra Ala-307-Asn….’
Line 432 phrase ‘and complicated for detection‘ was replaced by ‘and their detection is difficult’ was added
Lines 433-434 sentence was re-write ‘The same authors indicate that for these strains resistance to amoxicillin clavulanate is doubtful.’
Line 453 word ‘reports’ was replaced by ‘highlights’
Line 455 word ‘assay’ was replaced by ‘study’
Line 456 phrase was re-write ‘These methods may be a useful tool for….’
Round 2
Reviewer 2 Report
since you have added some new parts in the text, I made several notifications for you

Author Response
We apologize to the Reviewer for our shortcomings unclear or missing wording, which confused and raised the Reviewer's doubts, and thank You very much for valuable tips and quick response to our comments. Your recommendations significantly improved the quality of our manuscript.
With reference to the Reviewer's comments, following changes have been added to the manuscript:
Answer 1: Line 19: sentence has been corrected „Beta-lactams are the most frequently used antimicrobials, and the……”
Answer 2: Lines 25-28: parentheses have been used for the percentages “(54.4%), followed by blaOXA (19.3%), blaDHA (12.3%), blaSHV (10.5%), blaGES (7.0%), blaCMY (5.3%), blaVEB (1.8%) and blaROB-1 (1.8%). Lys-526 was the most common substitution in ftsI gene. The resistance genotypes were as follows: gBLNAS (17.5%), low-gBLNAR I (1.8%), low-gBLNAR II (1.8%), gBLNAR II (15.8%), gBLPAS (15.8%), gBLPAR (19.3%), gBLPBS I (8.8%) and gBLPBS II (1.8%);”
Answer 3: Line 27: phrase “useful a tool” has been corrected
Answer 4 : Line 33: sentence has been corrected “Beta-lactam antibiotics and especially penicillins are the most frequently used group of antimicrobials in the European Union”
Answer 5 : Lines 46-48: sentence has been corrected “Due to the very frequent and widespread use of beta-lactams, the development of resistance to this group of drugs has been observed also among isolates of H. influenzae and Haemophilus parainfluenzae”
Answer 6 : Line 50: phrase has been corrected “and PBP3 are the most correlated’
Answer 7 : Line 58: phrase has been corrected “Both belong to Ambler’s class A’
Answert 8: Lines 68-70: sentence has been corrected “According to literature, the third strategy is possible consists of the simultaneous presence of both mechanisms of resistance in the same strain, so that this strain can produce beta-lactamases and as well as present modifications in the structure of PBP proteins”
Answer 9 : Line 73: phrase has been corrected “To achieve this,….”
Answer 10:Line 112: word ‘and’ has been added to the phrase “As for ampicillin and amoxicillin”
Answer 11 :Line 115: phrase has been added “the isolate should be reported”
Answer 12 :Lines 113 and 117: phrase “the isolate should be reported as” has been corrected
Answer 13 :Line 164: parentheses for table 2 have been used
Answer 14 :Lines 173-176: parentheses have been used “Sensitivity (defined as the percentage of true positives) was calculated as the proportion of H. parainfluenzae isolates with positive test among all isolates tested which actually produced beta-lactamase, whereas specificity (defined as the percentage of true negatives) was calculated as….”
Answer 15 :Line 176: phrase has been corrected “the proportion of isolates with negative test”
Answer 16 :Line 179: word ‘value’ has been added to the sentence “Positive predictive value was the probability….’
Answer 17 :Lines 203-206: sentence has been re-write: ‘According to result of beta-lactamase synthesis (on the basis of three phenotypical methods as follows: cefinase test, penicillinase production in API NH microtest and amoxicillin-clavulanate 2/1 µg susceptibility test) altogether,…’
Answer 18 : Line 208: phrase “gave uncertain or ambiguous result” has been re-written
Answer 19 :Line 209: word ‘which’ has been added
Answer 20 :Lines 230, 231, 247, 348 and 399: phrase “beta-lactam resistant” have been used
Answer 21: Line 273: phrase has been corrected “due to the presence of amino-acid substitution”
Answer 22 :Lines 275-277: sentence has been re-written: “Similarly, two cefinase-positive, one resistant to four (Am AmC Cxm Ctx) and one resistant to six (Am AmC Cxm Ctx Ipm Mem) beta-lactam antibiotics BLPBR isolates, with Lys-526 substitution in ftsI gene, were classified into gBLPACR I genotype.”
Answer 23 :Line 281: phrase has been corrected “etiological factor of such infections as caused by H. parainfluenzae”
Answer 24Line 282: phrase has been corrected “have a huge impact”
Answer 25 Line 283: a comma has been added to the phrase “The importance of this issue is, among….”
Answer 26 Lines 286-288: sentences has been linked and re-written “Since 2013 in PubMed database only a few publications exist about H. parainfluenzae resistance to beta-lactam antibiotics and their molecular mechanisms [7,13,16,21], despite the fact that it belongs to the group of fastidious Gram-negative bacteria of Haemophilus - Aggregatibacter - Cardiobacterium - Eikenella - Kingella genus (HACEK). That means H. parainfluenzae is an….”
Answer 27 Line 302: phrase has been corrected “resistance to ampicillin, cefuroxime and cefotaxime predominating…..”
Answer 28 Line 312: the numbers have been ordered in the phrase “to five, six or eight drugs from different therapeutic groups”
Answer 29 Line 321: sentence has been corrected “When screening is negative, all beta-lactam agents for which clinical…….”
Answer 30 Line 330: phrase has been corrected “and in 4.6% isolates at least one of the used”
Answer 31 Line 331: phrase has been corrected “gave a less precise and difficult to interpret (weak colour change) result”
